



# Hydroxyl Radical in/on Illuminated Polar Snow: Formation Rates, Lifetimes, and Steady-State Concentrations

Zeyuan Chen[1], Liang Chu[1], Edward S. Galbavy[1,2], Keren Ram[1], Cort Anastasio[1]

[1]Department of Land, Air, and Water Resources, University of California, Davis, CA, 95616, USA.

[2]Now at NOVA Engineering and Environmental, Panama City Beach, Fl, 32408, USA.

*Correspondence to*: Cort Anastasio (canastasio@ucdavis.edu)

Keywords: ice photochemistry, snowpack, oxidants, organic carbon, reactive halogens

**Abstract.** While hydroxyl radical (˙OH) in the snowpack is likely a dominant oxidant for organic

species and bromide, little is known about the kinetics or steady-state concentrations of ˙OH on/in snow and ice. Here we measure the formation rate, lifetime, and concentration of ˙OH for illuminated polar snow samples studied in the laboratory and in the field. Laboratory studies show that ˙OH kinetics and steady-state concentrations are essentially the same for a given sample studied as ice and liquid, in contrast to other oxidants, which show a concentration enhancement in ice relative to solution. The

average production rate of ˙OH in samples studied at Summit, Greenland is 5 times lower than the average measured in the laboratory, while the average ˙OH lifetime determined in the field is 5 times higher than in the laboratory. These differences indicate the polar snows studied in the laboratory are affected by contamination, despite efforts to prevent this. Steady-state concentrations of ˙OH in snow studied in the field at Summit, Greenland range from $(0.8 \text{ to } 3) \times 10^{-15}$ M, comparable to values

reported for mid-latitude cloud and fog drops, rain, and deliquesced marine particles, even though impurity concentrations in the snow samples are much lower. Partitioning of firn-air ˙OH to the snow grains will approximately double the steady-state concentration of snow-grain hydroxyl radical, leading to an average [˙OH] in near-surface, summer Summit snow of approximately $4 \times 10^{-15}$ M. At this concentration, the ˙OH-mediated lifetimes of organics and bromide in Summit snow grains are

approximately 3 days and 7 hours, respectively, suggesting that hydroxyl radical is a major oxidant for both species.



## 1 Introduction

Hydroxyl radicals ($^{\bullet}$OH) are ubiquitous in the atmosphere and react readily with most organic compounds; as such, they play an important role in chemical processing and in controlling the oxidizing capacity of the troposphere (Grannas et al., 2007b; Thompson and Stewart, 1991). Although $^{\bullet}$OH has been studied in atmospheric and terrestrial waters (including cloud and fog drops and surface waters) (Arakaki et al., 2013; Ashton et al., 1995; Anastasio and McGregor, 2001; Herrmann et al., 2010), little is known of $^{\bullet}$OH in snowpacks (Galbavy et al., 2007; Beyersdorf et al., 2007).

Snowpacks have two general components: the solid snow grains (and their associated impurities) and the interstitial (firn) air (Bartels-Rausch et al., 2014). From hydrocarbon decay measurements, $^{\bullet}$OH concentrations in the near-surface firn air at Summit, Greenland peak at $1.5 \times 10^6$ molecules cm$^{-3}$ in the spring (close to the ambient air value) and $3.2 \times 10^6$ molecules cm$^{-3}$ in July (approximately 20 – 30% lower than ambient) (Beyersdorf et al., 2007). There are no measurements of $^{\bullet}$OH concentrations in/on snow grains, but some is known about the production rate of $^{\bullet}$OH ($P_{\text{OH}}$). Based on laboratory determinations of the quantum yields for $^{\bullet}$OH formation from photolysis of hydrogen peroxide (HOOH), nitrate (NO$_3^-$) and nitrite (NO$_2^-$), estimated values of $P_{\text{OH}}$ in polar surface snow during summer are on the order of 100 - 300 nM hr$^{-1}$, with nearly all from photolysis of HOOH (Chu and Anastasio, 2005; France et al., 2007; Chu and Anastasio, 2003). The only measurements of $^{\bullet}$OH formation rates in/on snow grains are at Summit, with summer values typically in the same range as estimated previously and HOOH accounting for 97% or more of photoformed $^{\bullet}$OH (Anastasio et al., 2007). While the sink for $^{\bullet}$OH has not been measured in snow grains, based on snow composition it appears that organic compounds are the dominant $^{\bullet}$OH sinks (Grannas et al., 2004; Anastasio et al., 2007). There are also no measurements of the $^{\bullet}$OH steady-state concentration in snow grains, which makes it difficult to estimate the importance of $^{\bullet}$OH as an oxidant for organics and bromide.

As discussed in these previous studies, $^{\bullet}$OH reactions in/on snow are likely an important sink for snowpack trace species, which will influence the lifetimes, toxicities, and transformations of these contaminants. For example, $^{\bullet}$OH reacts with snow-grain organic matter to form volatile organic compounds (VOCs) such as formaldehyde, which can be released to the atmosphere (Grannas et al., 2004; Anastasio and Jordan, 2004; Domine and Shepson, 2002; Jacobi et al., 2006; Anastasio et al.,





2007). Oxidation by ˙OH can also convert snowpack halides (especially Br⁻) into reactive volatile halogens, such as $Br_2$, which can alter ozone and hydrocarbon chemistry in both the snow interstitial air and the atmospheric boundary layer (Grannas et al., 2007b; Wren et al., 2013; Abbatt et al., 2010; Anastasio et al., 2007; Chu and Anastasio, 2005; Pratt et al., 2013; Thomas et al., 2011). Additionally,
˙OH might alter ice core records of past atmospheres by reacting with trace species in snow (Anastasio and Jordan, 2004).

Recent work has shown that the steady-state concentrations of some oxidants can be much higher in/on illuminated ice compared to in the same sample studied as solution (Bower and Anastasio, 2013b; Bower and Anastasio, 2013a). For example, the singlet oxygen concentration in ice can be higher by a
factor of approximately 10,000 (Bower and Anastasio, 2013b) while the concentration of a triplet excited state can increase by a factor of roughly 100 (Chen and Anastasio, In preparation). These enhancements occur because of the freeze-concentration effect, where solutes are excluded to liquid-like regions (LLRs) in/on the ice, resulting in much higher effective concentrations in these small domains (Grannas et al., 2007a; Cho et al., 2002; Bower and Anastasio, 2013b; Bower and Anastasio,
2013a). While it has0 not been measured experimentally, the freeze-concentration effect might also alter ˙OH concentrations within snow and ice (compared to the corresponding solution), which would alter its impacts. Our goals in this work are to: (1) measure the steady-state concentration of ˙OH, and the ˙OH kinetics (i.e., its rate of formation and lifetime), in polar snow samples, (2) compare ˙OH measurements in samples studied in the laboratory and in the field, and (3) examine how ˙OH kinetics and
concentrations vary between solution and ice.  To achieve these goals we used a benzoate probe technique to characterize photoformed ˙OH in polar snow samples studied in the lab (as solution and ice) and in the field (studied as ice).

## 2 Experimental Methods

### 2.1 Materials

Hydrogen peroxide and acetonitrile (Optima) were from Fisher, sodium benzoate (99%) and 2-nitrobenzaldehyde (2NB; 98%) were from Sigma-Aldrich, and *p*-hydroxybenzoate (*p*-HBA; 98%) was





from TCI America. All chemicals were used as received. Purified water ("Milli-Q water") was obtained from a Milli-Q Plus system ($\geq$ 18.2 M$\Omega$ cm) with an upstream Barnstead B-Pure cartridge to remove organics.

## 2.2 Snow sample collection

Surface snow samples (approximately 0 – 3 cm depth) were collected from undisturbed areas within the clean air sectors at Summit, Greenland (72.6°N, 38.5°W, 3200 m elevation) and Dome C, Antarctica (75.1°S, 123.4°E, 3270 m elevation). For the 2005 Summit samples, snow was removed with polytetrafluoroethylene (PTFE) instruments, set on a 3 ft × 3 ft Teflon sheet, mixed, and placed in glass Schott bottles (100 or 250 mL). In 2006 and 2007, samples were collected directly into the Schott

bottles. See Supplemental Table S1 for sampling conditions. Samples to be studied in the laboratory were held in cold, dark storage at Summit and Dome C and shipped frozen to UC Davis, where they were stored in a freezer (– 20 °C) for 2 to 26 months prior to being studied. Samples studied in the field at Summit were used within a week of collection. The field blank for Summit was prepared using Milli-Q from the field that was frozen in Schott bottles and shipped back to UC Davis with the samples. The

Dome C field blank was Milli-Q water shipped from our laboratory, frozen at Dome C, and then shipped back with the samples.

## 2.3 Laboratory sample preparation

Each laboratory sample was studied twice: once to determine the formation rate of $^{\bullet}$OH ($P_{OH}$) and once to determine the apparent rate constant for $^{\bullet}$OH destruction ($k'_{OH}$). For each type of test, the snow

sample was first melted overnight in the refrigerator. To determine $P_{OH}$, 200 μM benzoate (BA) was added to a portion of the melted sample to scavenge essentially all of the $^{\bullet}$OH formed during illumination. For measuring $k'_{OH}$, we took four aliquots of the sample, added 100 μM HOOH to each (to increase the rate of $^{\bullet}$OH formation and better determine the $^{\bullet}$OH sink), and added a different BA concentration (typically between 20 and 200 μM) to each. Samples studied as ice were frozen in a

covered, custom-built, Peltier-cooled freeze chamber at -10 °C (Bower and Anastasio, 2013b).



Laboratory blanks were prepared in the same way as the samples but with fresh Milli-Q instead of melted snow.

### 2.4 Laboratory container types and cleaning treatments

We tested three types of sample containers in the lab to explore which would minimize contamination:
(1) 1-mL white PTFE Teflon beakers (15 mm H, 8 mm ID, Fisher Scientific), (2) 4-mL rectangular quartz cells (1-cm path length) with air-tight screw caps and Teflon septa (FUV, Spectrocell), and (3) 400-µL quartz tubes (30 mm L, 5 mm ID, 1 mm wall thickness) custom-made from GE 021 quartz and sealed with white silicone caps. We also explored two methods of extra cleaning for the containers after completing our normal cleaning procedures: (1) add Milli-Q to the container and illuminate for 24 hours with 254 nm radiation in an RPR-100 photoreactor equipped with 16 mercury lamps (25 W) (Southern New England Ultraviolet Company), and (2) in the same way but with 100 µM HOOH added to the Milli-Q prior to illumination as a photochemical source of $^\bullet$OH.

### 2.5 Laboratory sample illumination and analysis

For laboratory determinations of $^\bullet$OH kinetics, we apportioned samples (with added BA or with BA and 100 µM HOOH; Section 2.3) into cleaned (HOOH + UV) quartz tubes with silicon caps. Samples were held in a custom designed, Peltier-cooled chamber illuminated with simulated sunlight (Ram and Anastasio, 2009). The solar simulator simulates total global solar radiation at a solar zenith angle of 48.2°. Prior to illumination, samples were allowed to thermally equilibrate in the illumination system chamber in the dark (30 min for liquid samples and 60 - 80 min for ice samples). During illumination $^\bullet$OH reacts with BA to form the stable product p-HBA. To determine the rate of p-HBA formation, 50 µL of sample (10 µL for rinse and 40 µL for injection) was removed at known times and p-HBA was measured using HPLC with UV-Vis detection (Anastasio et al., 2007). For ice samples, the entire tube was removed for a given time point. We used the same procedures for laboratory and field samples. Dark control samples were run in parallel in the illumination system chamber, using the same sample and container conditions except that the tube and cap were covered with aluminum foil. Lastly, we measured the photon flux ($I_\lambda$) in all samples by measuring the direct photodegradation of 4 µM of 2NB



($j_{2NB}$) in the same type of container and same temperature as that in the sample measurement (Chu and Anastasio, 2003).

## 2.6 Field studies

For experiments at Summit, the snow sample in the sealed Schott bottle was left in a heated building (~
10 °C) in the dark to melt during the day. That evening we divided the sample into several portions, added a different concentration of BA (2 – 11 μM) to each, and created ice pellets by freezing 1-mL aliquots of sample on a PTFE sheet within a trench dug below the Summit Science Lab. (Note that we had planned to add 10 times more BA to the samples, but mistakenly made a lower concentration BA stock solution that was not discovered until after the field campaign ended.) After freezing we kept the
pellets outdoors (below freezing) in 30-mL amber glass jars wrapped in aluminum foil until the moment of exposure. To start illumination, we placed the pellets on the snow surface, and quickly picked up an ice pellet as the time zero point (Anastasio et al., 2007). Samples were then collected at known times, put into sealed jars, melted in the dark, and analyzed for *p*-HBA and BA using HPLC. For dark samples we used the same pellets, but placed under a tub next to the sunlit samples to eliminate illumination. We
also simultaneously illuminated blank controls, which were ice pellets made with Milli-Q water (and BA) instead of snow.

## 2.7 •OH kinetic analysis

In laboratory samples used to determine the formation rate of •OH (i.e., with 200 μM BA added), we expect essentially all of •OH will react with BA. Under this condition the experimentally measured rate
of •OH formation ($P_{OH,exp}$) is determined as

$$P_{OH,exp} = P_{p-HBA}/Y_{p-HBA} \tag{1}$$

where $P_{p-HBA}$ is the rate of *p*-HBA formation, determined from the slope of a linear regression of [*p*-HBA] versus illumination time, and $Y_{p-HBA}$ is the yield of *p*-HBA from the •OH reaction with BA (0.19 ± 0.0068 and 0.081 ± 0.014 for solution and ice, respectively (Anastasio and McGregor, 2001; Chu and
Anastasio, 2003)).

In laboratory samples with 100 μM added HOOH (and different amounts of BA), we determined





the apparent rate constant for $^{\bullet}$OH destruction, $k'_{OH}$, which is the inverse of the $^{\bullet}$OH lifetime ($\tau_{OH}$). This pseudo-first-order rate constant reflects the concentrations and reactivities of all of the $^{\bullet}$OH sinks in each sample and is equal to the product of the second-order rate constant for each sink $i$ and its concentration, $[i]$, summed over all of the sinks, i.e., $k'_{OH} = \Sigma k_{OH+i}[i]$. The value of $k'_{OH}$ for each

sample is determined from a linear regression of $1/P_{p-HBA}$ versus $1/[BA]$ (i.e., the "inverse" plot) (Zhou and Mopper, 1990; Anastasio and McGregor, 2001):

$$k'_{OH} = k_{BA+\cdot OH} \times (slope \, / \, y-intercept) \tag{2}$$

where $k_{BA+\cdot OH}$ is the second-order rate constant for reaction of BA and $^{\bullet}$OH (in M$^{-1}$s$^{-1}$), estimated from $k_{BA+\cdot OH} = \exp(26.6 - (1194.8/T))$ for both solution and ice samples (Ashton et al., 1995). We

subtracted the $k'_{OH}$ contribution from the 100 μM HOOH added to each sample; this value is $4.5 \times 10^3$ M$^{-1}$ s$^{-1}$ based on a 2$^{nd}$-order rate constant (Dorfman, 1973) for $^{\bullet}$OH and HOOH of $4.5 \times 10^7$ M$^{-1}$ s$^{-1}$. Lastly, we determined the steady-state concentration of $^{\bullet}$OH by combining the measured production rate (from the sample without added HOOH) and measured $k'_{OH}$,

$$[\cdot OH]_{exp} = \frac{P_{OH,exp}}{k'_{OH}} \tag{3}$$

The experimentally determined values of $P_{OH,exp}$ and $[\cdot OH]_{exp}$ were then normalized to give the values expected under midday, summer solstice sunlight conditions at Summit:

$$P_{OH,Sum} = P_{OH,exp} \times \frac{j_{2NB,Sum}}{j_{2NB,exp}} \tag{4}$$

$$[\cdot OH]_{Sum} = [\cdot OH]_{exp} \times \frac{j_{2NB,Sum}}{j_{2NB,exp}} \tag{5}$$

where $j_{2NB,Sum}$ is the rate constant for loss of the actinometer 2NB under midday, summer solstice

sunlight at Summit (0.02 s$^{-1}$) (Galbavy et al., 2010) and $j_{2NB,exp}$ is the value measured on the day of a given laboratory experiment.

For field samples, we determined $P_{OH,Sum}$, $k'_{OH}$ (Eq. 2), and $[\cdot OH]_{Sum}$ using the "inverse" plot from the ice pellets studied with a range of BA concentrations:





$$P_{\text{OH,Sum}} = (y - \text{intercept} \times Y_{p-HBA})^{-1} \qquad (6)$$

$$[\cdot\, \text{OH}]_{\text{Sum}} = \left(k_{BA+\cdot OH} \times \text{slope} \times Y_{p-HBA}\right)^{-1} \qquad (7)$$

In contrast to our past work (Anastasio et al., 2007), [BA] in the ice pellets was stable and therefore was not needed to normalize $p$-HBA concentrations. Because the rates of formation and steady-state concentrations for the field samples were measured using ambient Summit sunlight, they were not normalized to $j_{2NB,Sum}$.

### 2.8 Calculated organic carbon concentration

Organic compounds are likely the dominant sink for $^{\bullet}$OH in/on snow grains. As shown by Arakaki et al., the second-order rate constant for $^{\bullet}$OH reaction with organic carbon (L (mol-C)$^{-1}$ s$^{-1}$) is very similar in different atmospheric waters and even in surface waters (Arakaki et al., 2013). These similarities indicate that the apparent $^{\bullet}$OH scavenging rate constant in environmental waters is primarily controlled by the organic carbon concentration and is relatively insensitive to differences in the complex mixtures of organic compounds in different samples. Thus we estimated the concentration of dissolved organic carbon in each sample, [DOC] (mol-C L$^{-1}$), using:

$$[\text{DOC}] \approx \frac{k'_{\text{OH}}}{k_{\text{C,OH}}} \qquad (8)$$

where $k'_{\text{OH}}$ is the measured pseudo-first-order $^{\bullet}$OH scavenging rate constant (s$^{-1}$) and $k_{\text{C,OH}}$ is the general bimolecular rate constant between $^{\bullet}$OH and organic carbon, determined as $(3.8 \pm 1.9) \times 10^{8}$ L (mol-C)$^{-1}$ s$^{-1}$ in atmospheric waters (Arakaki et al., 2013).

### 3 Results and Discussion

### 3.1 Minimizing background contamination for $^{\bullet}$OH sinks

To examine whether our containers or handling might add $^{\bullet}$OH-scavenging contaminants to our samples, we first performed a series of tests on laboratory Milli-Q blanks in three different types of containers (quartz cells, Teflon beakers, and quartz tubes) with three types of additional cleaning





methods (no additional cleaning, 254-nm UV treatment, HOOH + UV treatment). These additional methods were added after finishing with our standard cleaning procedures, which are described below. After the additional cleaning, fresh Milli-Q and 100 μM HOOH was added to each container, BA was added at a range of concentrations to different sample aliquots, and the samples were illuminated in order to determine $k'_{OH}$.

The first container type we examined was a 1-cm rectangular, air-tight quartz cell, where the standard cleaning is copious Mill-Q rinsing before and after a rinsing with a 50:50 MeOH:1 M $H_2SO_4$ solution. As shown in Figure 1, without any additional cleaning, $k'_{OH}$ at 293 K is $1.1 \times 10^5$ s$^{-1}$, which is quite high and corresponds to an estimated organic carbon concentration of 300 μmol-C L$^{-1}$ based on Eq. (8). UV irradiation decreases this background by approximately a factor of two, while the combination of HOOH + UV treatment decreases the background by a factor of 10 (Figure 1). However, because the expensive quartz cells had a tendency to crack upon sample freezing, we next tested Teflon beakers as a container.

Ice and solution Milli-Q blanks in Teflon beakers have $k'_{OH}$ values of $(1 - 1.7) \times 10^5$ s$^{-1}$ using only our standard cleaning condition, which is Alconox wash, Milli-Q rinse, ethanol rinse, Milli-Q rinse, and then an overnight Milli-Q soak. In contrast, additional HOOH + UV treatment can produce $^{\bullet}$OH destruction apparent rate constants as low as seen in the quartz cells ($8 \times 10^3$ s$^{-1}$). However, HOOH + UV treated Milli-Q blanks studied as ice were approximately $5 - 10$ times higher, suggesting that contaminant gases can adsorb onto the beaker samples and that a closed container is needed.

Therefore, we tested homemade quartz tubes with silicon caps as our third container type. The standard cleaning procedure for these tubes is the same as the Teflon beakers. Figure 1 shows that with additional UV + HOOH treatment the quartz tubes can achieve $^{\bullet}$OH rate constants as low as the other containers, corresponding to organic carbon concentrations as low as 20 μmol-C L$^{-1}$. Since the quartz tubes with HOOH + UV treatment gave low amounts of background $^{\bullet}$OH scavengers, were inexpensive, and only occasionally broke upon freezing, we conducted all laboratory measurements of $P_{OH}$ and $k'_{OH}$ in snow samples using these containers.





### 3.2 $P_{OH,Sum}$, $k'_{OH}$, and $[{}^{\bullet}OH]_{Sum}$ in laboratory illuminated samples

Figure 2 shows the results of laboratory experiments conducted on one of the snow samples (Summit 0526). Figure 2a illustrates that the rate of $p$-HBA formation increases with increasing [BA] due to [BA] intercepting a larger fraction of the photoformed ${}^{\bullet}OH$ compared to the natural ${}^{\bullet}OH$ scavengers in

the sample. The competition between [BA] and other scavengers can be seen more clearly in Figure 2b where $P_{p\text{-HBA}}$ increases as the [BA] concentration increases and plateaus at BA concentrations of 50 μM and higher, which indicates BA is scavenging most OH at and above this concentration. Thus essentially all ${}^{\bullet}OH$ will be scavenged by BA in our samples studied in the laboratory to measure $P_{OH,exp}$, where we used 200 μM of BA. Lastly, Figure 2c shows an example of the "inverse" plot used to find

$k'_{OH}$ (Eq. 2) in the laboratory samples, which is then combined with $P_{OH,Sum}$ (Eq. 4) to determine $[{}^{\bullet}OH]$ (Eqs. 3 and 5).

Figure 3 shows values of $P_{OH,Sum}$ measured at 263 and 274 K for three different Summit samples, one Dome C sample, field blanks from Summit and Dome C, and a laboratory blank. Values of $P_{OH,Sum}$ are similar at Summit and Dome C (although there is only one Dome C sample) for ice samples (263 K)

and range from approximately 330 – 840 nM hr⁻¹, which is somewhat higher than past summer results measured at Summit (130 to 610 nM hr⁻¹) (Anastasio et al., 2007) and higher than the calculated rate of ${}^{\bullet}OH$ formation on snow grains from HOOH photolysis (approximately 250 nM hr⁻¹) (Chu and Anastasio, 2005). Solution values (274 K) of $P_{OH,Sum}$ are similar to their corresponding ice values but generally slightly higher; this is likely due in part to the temperature dependence of the ${}^{\bullet}OH$ quantum

yield from HOOH photolysis, which is 10% higher at 274 K than at 263 K (Chu and Anastasio, 2005). Blank rates range from 80 to 160 nM hr⁻¹ and represent 10 to 40% of the corresponding sample values.

Figure 4 depicts the apparent rate constant for ${}^{\bullet}OH$ destruction ($k'_{OH}$) and its inverse – the lifetime of ${}^{\bullet}OH$ ($\tau_{OH}$) – for the same set of samples shown in Figure 3. As was seen for $P_{OH}$, rate constants for ${}^{\bullet}OH$ destruction ($k'_{OH}$) are similar between solution and ice for a given sample and are in the same range for

the Summit and Dome C samples. On average, $k'_{OH}$ in solution at 274 K is 20% higher than the ice value at 263 K. This small difference is probably due to the temperature dependence of the ${}^{\bullet}OH$ bimolecular rate constants with organic scavengers. For example, $k_{BA+{}^{\bullet}OH}$ at 274 K is 20% higher than the value at 263 K; many of the natural scavengers probably have similar temperature dependence.





$k'_{OH}$ values for the samples without blank correction are in the range of $(3.0 - 7.8) \times 10^4$ s$^{-1}$, while blank values range from $(0.83 - 2.8) \times 10^4$ s$^{-1}$ (Figure 4). The average of the blank values is roughly 10, 30, and 60% of the corresponding average sample values at 293, 274, and 263 K, respectively. This suggests that the sample values without blank correction are upper bounds of the true values. We have

not corrected the sample values for the blanks because the blank levels might or might not reflect contamination in the samples. For example, if the blank values are the result of low levels of contamination in the Milli-Q water then this will not affect the samples. On the other hand, if the blank values result from low levels of organics in the illumination containers, then the samples would be expected to have the same background level.

Using results from Figure 3 and 4, we can combine $P_{OH,Sum}$ and $k'_{OH}$ to calculate [$^{\bullet}$OH]$_{Sum}$ (Eq. 7). As shown in Figure 5, steady-state $^{\bullet}$OH concentrations normalized to midday, Summit summer solstice sunlight are $(2 - 5) \times 10^{-15}$ M and are similar between ice and solution. These snow concentrations of $^{\bullet}$OH are comparable to average values in mid-latitude cloud and fog drops, marine particles, and rain, which are generally $(0.5 - 7) \times 10^{-15}$ M (Arakaki et al., 2013; Anastasio and McGregor, 2001). This is

somewhat surprising since the polar snow samples are much cleaner than the mid-latitude drops and particles, which typically have orders of magnitude higher concentrations of contaminants; for example, organic carbon concentrations in Davis, CA fog waters (Anastasio and McGregor, 2001) are approximately 100 times higher than levels in polar snow (see below). In fact, $P_{OH,Sum}$ and $k'_{OH}$ are both much smaller in the snow samples (expressed relative to the melted snow volume) than in atmospheric

hydrometeors. However, since [$^{\bullet}$OH] is equal to the ratio of these parameters (Eq. 7), its value is very similar across mid-latitude drops and particles (Arakaki et al., 2013) as well as for the snow samples here.

This correlation between $^{\bullet}$OH sources and sinks is also responsible for the very similar concentrations of $^{\bullet}$OH between the solution and ice results for a given sample (Figure 5). The $^{\bullet}$OH

concentration is determined by the balance between its rate of formation and its pseudo-first-order rate constant for loss:

$$[\cdot OH] = \frac{P_{OH}}{k'_{OH}} \qquad\qquad\qquad (9)$$



For the melted snow samples this is approximately equal to

$$[\cdot OH]_{LIQ} \approx \frac{j(HOOH \rightarrow \cdot OH)[HOOH]_{LIQ}}{\sum k_{i+\cdot OH}[i]_{LIQ}} \qquad (10)$$

where $j(HOOH \rightarrow \cdot OH)$ is the rate constant for $^{\bullet}OH$ formation from HOOH photolysis and $i$ represents natural (likely organic) $^{\bullet}OH$ scavengers. Upon freezing the snow solution, solutes should be mostly

excluded to liquid-like regions containing high concentrations of solutes (Bartels-Rausch et al., 2014); for example, at 263 K the total solute concentration in LLRs is predicted to be 5.4 M (Cho et al., 2002), which represents a freeze-concentration factor of approximately 50,000 for a snow sample with an initial (melted) total solute concentration of 100 μM. As a result of this enhancement in solute concentrations, both $P_{OH}$ and $k'_{OH}$ in the LLRs should be approximately 50,000 times higher than in the

melted sample. The resulting expression for [$^{\bullet}OH$] in the LLRs is

$$[\cdot OH]_{LLR} \approx \frac{j(HOOH \rightarrow \cdot OH)[HOOH]_{LLR}}{\sum k_{i+\cdot OH}[i]_{LLR}} \qquad (11)$$

Since the LLR concentration of a given solute is larger than the initial solution concentration by approximately the freeze-concentration factor ($F$), we can re-express Eq. (11) as

$$[\cdot OH]_{LLR} \approx \frac{j(HOOH \rightarrow \cdot OH)[HOOH]_{LIQ}F}{\sum k_{i+\cdot OH}[i]_{LIQ}F} \qquad (12)$$

The similarities in the ice and solution results in Figure 5 indicate that the freeze-concentration factor is essentially the same for both the $^{\bullet}OH$ sources and sinks. Thus, the freeze-concentration factor approximately cancels in Eq. (12) and the steady-state concentration for $^{\bullet}OH$ in LLRs of ice is essentially the same as in solution. In contrast, in the case of $^{1}O_{2}^{*}$, the source of singlet oxygen is enhanced by the freeze-concentration factor but the sink (liquid $H_2O$) is essentially the same in LLRs

and solution (Bower and Anastasio, 2013a; Bower and Anastasio, 2013b). The result is that [$^{1}O_{2}^{*}$] in ice LLRs is enhanced by the freeze concentration factor, which is on the order of $10^{4}$ or higher for typical polar snows (Bower and Anastasio, 2013b).

We can also use equations 9 – 12 to gain a better understanding of kinetics in the liquid-like regions. While Figures 3 and 4 show that $P_{OH}$ and $k'_{OH}$ are essentially the same for a sample studied as solution

or ice, this similarity is somewhat misleading. As shown in the numerator of equation 12, the rate of





$^{•}$OH production in the LLRs should be $F$ times higher than the solution rate. However, since we melt the sample to analyze it, we cannot determine the concentration of photoproduced $p$-HBA in the LLR volume, but only in the melted sample volume, which is higher than the LLR volume by a factor of $F$. The same should be true for the apparent rate constant for $^{•}$OH loss: as shown in the denominator of

equation 12, $k'_{OH}$ in the LLR should be $F$ times higher than the solution value, but we do not see this since can only do our analysis in the melted sample. Thus, on a whole (melted) sample volume basis, $P_{OH}$ and $k'_{OH}$ are the same in solution and ice, but in the native LLR volume, both the rate of $^{•}$OH formation and apparent rate constant for $^{•}$OH loss are much higher than in solution. On the other hand, [$^{•}$OH] is essentially the same in both ice and solution, regardless of whether we consider LLR or total

melted sample volume, since it is the ratio of $P_{OH}$ to $k'_{OH}$.

### 3.3 $P_{OH}$, $k'_{OH}$, and [$^{•}$OH] in the field

In addition to the Summit and Dome C samples that we studied in the laboratory, we also studied a number of snow samples in the field at Summit. Of the 38 field samples (Figure 6), most were studied

only at one BA concentration, but four (experiments 207, 225, 254 and 263) used 2 to 4 BA concentrations and thus can be used to derive $P_{OH}$, $k'_{OH}$, and [$^{•}$OH]. As can be seen from Table 1, $P_{OH}$ values (7 to 200 nM hr$^{-1}$) for these four samples studied in the field are, on average, 5 times lower than values for samples studied in the laboratory as ice (350 to 600 nM hr$^{-1}$). While these are different sets of samples, this large difference indicates that the samples studied in the laboratory were contaminated,

either from shipping, storage, or handling during the experiments. We can also estimate $^{•}$OH production rates for the 34 field samples that were studied at only one BA concentration (i.e., the green diamonds in Figure 6). Using the average of the slopes from the four experiments with multiple BA concentrations in Figure 6 (644 hr µM$^{-1}$ µM), we can extrapolate each green diamond in Figure 6 to its corresponding y-intercept, and then use Eq. 6 to estimate $P_{OH}$. The resulting $^{•}$OH formation rates range

from 9 to 370 nM hr$^{-1}$, with an average (± σ) value of 100 ± 90 nM hr$^{-1}$ ($n = 34$); these values are comparable to $P_{OH}$ values for the four field samples studied with multiple BA concentrations (Table 1).

Similar to the field-lab relationship in $P_{OH}$, $k'_{OH}$ measured in the field, $(0.2 - 1) \times 10^4$ s$^{-1}$, is, on average, 5 times lower than values measured in the laboratory ice samples, $(3.0 - 6.2) \times 10^4$ s$^{-1}$. These

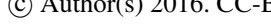



large differences in $P_{OH}$ and $k'_{OH}$ between samples studied in the lab and field indicate that transport to Davis, storage, and/or handling in the laboratory added significant amounts of contaminants that are both sources and sinks for $^\bullet$OH. This is also apparent from laboratory measurements of $k'_{OH}$ for the blanks, which are only somewhat smaller than the sample results (Figure 4). In contrast to the $P_{OH}$ and

$k'_{OH}$ results, values of [$^\bullet$OH] measured in field are quite similar to those determined in the laboratory, with average values of $2 \times 10^{-15}$ M and $3 \times 10^{-15}$ M, respectively. However, [$^\bullet$OH] is only close between the two sets of samples because the contamination in both $P_{OH}$ and $k'_{OH}$ essentially cancels in the calculation of the steady-state $^\bullet$OH concentration (Eq. 9).

### 3.4 Organic carbon in samples

We next use our measured values of $k'_{OH}$ to estimate concentrations of organic carbon in our samples. As shown in Table S4, organic carbon concentrations in snow samples studied in the laboratory range from 90 to 190 μmol-C/L while the field and lab blanks (Milli-Q) range from 20 to 100 μmol-C/L. In contrast, estimated organic carbon concentrations in the field samples are 5 to 30 μmol-C/L (Table 1); these much lower DOC values for the field samples are consistent with the idea that there was

significant contamination in the samples studied in the lab. Compared to previously reported OC values, the snow organic carbon concentrations estimated from our field $k'_{OH}$ values are at the lower end of previous results for snow from Summit and Alert and are similar to snow and sea ice values from Barrow (Table 2). In comparison, concentrations in ice core samples tend to be significantly lower, indicating significant mineralization of the organic carbon after snow deposition.

### 3.5 Implications and Uncertainties

In addition to photolysis of HOOH and other chromophores in/on snow grains, transport of firn air $^\bullet$OH is another source of hydroxyl radical to snow grains. Based on firn air concentrations of $^\bullet$OH measuredat Summit (Beyersdorf et al., 2007), we previously estimated that the rate of mass transport of firn air $^\bullet$OH to the snow grains is roughly equal to the rate of $^\bullet$OH photoformation in/on the snow grains

(Chu and Anastasio, 2005). Thus, accounting for gas-to-grain partitioning of $^\bullet$OH will approximately double our measured values of $P_{OH,Sum}$ and [$^\bullet$OH]$_{Sum}$: based on our field results, the resulting snow-



grain steady-state concentrations of $^\bullet$OH at 263 K are $(2 - 6) \times 10^{-15}$ M, with an average of $4 \times 10^{-15}$ M. Based on $2^{nd}$-order rate constants of $^\bullet$OH with dissolved organic compounds (typically $1 \times 10^9$ M$^{-1}$ s$^{-1}$) (Ross, 1988) and bromide ($1.06 \times 10^{10}$ M$^{-1}$ s$^{-1}$) (Zehavi and Rabani, 1972), we estimate OC and Br$^-$ lifetimes as 3 days and 7 hours, respectively. This suggests that snow-grain $^\bullet$OH is significant in the

transformation of both snowpack organics and bromide, resulting in emissions of both volatile organic compounds and reactive halogen gases (Anastasio et al., 2007).

There are several important uncertainties in our results. Perhaps the most important is the uncertain impact of melting the snow samples and refreezing them as ice pellets (which is required in order to add the $^\bullet$OH probe). This likely alters the partitioning of solutes, e.g., possibly moving hydrogen peroxide

that is trapped in the bulk ice matrix into liquid-like layers. Given that HOOH is the dominant source of snow-grain $^\bullet$OH (accounting for more than 97% of $^\bullet$OH formed) (Anastasio et al., 2007; Chu and Anastasio, 2005), its movement from bulk ice to LLRs during sample preparation might increase the rate constant for HOOH photolysis (and thus the steady-state concentration of $^\bullet$OH) by roughly a factor of 2 or 3 based on measurements of HOOH photolysis in flash-frozen ices (Beine et al., 2012). Melting

and refreezing the snow might also affect the mixing state of solutes, possibly moving initially separate species together or possibly having the opposite effect. The potential impact of this movement is highly uncertain. While our results are the first experimental determinations of the $^\bullet$OH sink and steady-state concentration in snow samples, we hope that future developments include snow-grain $^\bullet$OH techniques that do not disturb snow morphology.

**4 Conclusion**

We have made the first complete measurements of $^\bullet$OH kinetics and steady-state concentrations in illuminated snow samples. For a given sample, we find that $^\bullet$OH concentrations are essentially the same whether the sample is studied as ice or liquid; the same is true for OH production rates and OH lifetimes,. This lack of enhancement on ice is different from what we have found for singlet molecular

oxygen and triplet excited states, both of which are enhanced by orders of magnitude in/on ice compared to in solution.

Production rates of $^\bullet$OH in samples studied at Summit during summer range between approximately



10 – 200 nM hr$^{-1}$, similar to previously measured and modeled rates. The lifetime of $^{\bullet}$OH in these samples is on the order of 70 to 500 µs, corresponding to organic carbon concentrations of 30 to 5 µmol-C L$^{-1}$. $^{\bullet}$OH concentrations in near-surface snow at Summit are $(2 – 6) \times 10^{-15}$ M, with approximately equal contributions from photolysis of snow-grain impurities (mostly HOOH) and

partitioning of $^{\bullet}$OH from the firn air. Compared to these samples studied in the field, samples studied in the laboratory show higher $^{\bullet}$OH production rates and lower $^{\bullet}$OH lifetimes, both as a result of contaminants. Similar contamination might be important for many laboratory studies of snow chemistry and should be characterized when possible. Based on our field measurements, hydroxyl radical in/on deposited snow grains and on atmospheric snow and ice likely plays an important role in the

transformation of organic compounds and bromide. These $^{\bullet}$OH-mediated reactions will release volatile organic compounds and reactive bromine species to the atmospheric boundary layer, leading to ozone depletion, mercury oxidation, and alterations to HO$_x$ chemistry.

*Author contributions.* L. Chu and C. Anastasio designed and carried out the laboratory sample

experiments. E S. Galbavy, K. Ram, and C. Anastasio designed and carried out the field sample experiments. Z. Chen and C. Anastasio wrote the manuscript with contributions from L. Chu.

*Acknowledgement.* The authors thank the National Science Foundation and the Greenland Home Rule Ministry of Environment and Nature for granting us permission to do research at Summit, Greenland.

We also thank Barry Lefer, Bernhard Stauffer, Karen Guldbaek Schmidt, and Manuel Hutterli for snow samples; VECO Polar for superb logistical support; and Ted Hullar for editorial help. Funding for this work was provided by the National Science Foundation (grants 0455055 and 1214121).

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

**Tables and Figure Captions**

**Figure 1.** Effects of container type and cleaning on the apparent rate constant for $^{\bullet}$OH destruction in Milli-Q blanks. Open bars represent containers with no treatment, slashed bars have UV treatment





(container with Milli-Q illuminated for ≥ 8 hours with 254 nm radiation), and solid bars are containers treated with 100 μM added HOOH followed by ≥ 8 hours of UV treatment. Bars within a given treatment are listed in chronological order, from the initial to the final experiment. Error bars represent ± 1 standard error, based on propagated errors of the linear regression of $1/p$-HBA vs. $1/[BA]$ and the

$2^{nd}$-order rate constant for $^{\bullet}OH$ with BA.

**Figure 2.** Example of kinetic results for Summit sample 0526 studied in the laboratory. Panel a) shows the increase in $p$-HBA concentration during illumination in 4 aliquots of the sample containing different initial BA concentrations ($[BA]_0$) illuminated for 30 min with simulated sunlight. The diamonds (blue),

squares (red), triangles (green), and circles (gray) correspond to BA concentrations of 20, 30, 50, and 200 μM, respectively. The lines represent linear regression fits to data at each BA concentration (with y-intercept fixed at zero). Panel b) shows the measured production rate of $p$-HBA for each BA concentration and the accompanying regression fit. Error bars represent ± 1 standard error based on the linear regression slopes in panel a. Panel c) is the "inverse" plot and linear regression fit, whose slope

and y-intercept are used to calculate $P_{OH}$, $k^{'}_{OH}$ and $[^{\bullet}OH]$ (see Sect. 2.7 for more details). Error bars based on ± 1 standard error calculated in panel b.

**Figure 3.** Rate of $^{\bullet}OH$ formation in Summit and Dome C snow samples studied as solution (274 K, red) and as ice (263 K, blue), after normalization to Summit summer solstice sunlight. Error bars represent ±

1 standard error, based on propagated errors in the rate of $p$-HBA formation ($P_{p\text{-}HBA}$) and $Y_{p\text{-}HBA}$. Sample values are not adjusted for $^{\bullet}OH$ formation in the blanks.

**Figure 4.** Laboratory-determined apparent rate constant for $^{\bullet}OH$ destruction (left axis) and the corresponding $^{\bullet}OH$ lifetime (right axis) in Summit and Dome C snow samples and Milli-Q field and lab

blanks (with 100 μM added HOOH) at 293 K (green), 274 K (red), and 263 K (blue). Error bars represent ± 1 standard error, based on propagated standard errors (see text for details).





**Figure 5.** Laboratory-determined steady-state ˙OH concentrations in Summit and Dome C snow samples at 274 K (red) and 263 K (blue). Error bars represent ± 1 standard error, based on propagated errors in the slope and y-intercept of the "inverse" plots.

5 **Figure 6.** Measurements of ˙OH in Summit snow samples studied in the field, including four samples studied with multiple BA concentrations (with experiment number listed). Green hollow diamonds represent the 34 other snow samples that were studied using just one BA concentration.





**Table 1.** Summary of snow samples studied in the field at Summit, Greenland.

| Date | Expt # | Illumination Time [a] | T Range (K) | $P_{OH}$ (nM/hr) | $k'_{OH}$ (s$^{-1}$) | $\tau_{OH}$ (µs) | [$^{\bullet}$OH] (10$^{-15}$M) | [DOC][b] (µM) |
|---|---|---|---|---|---|---|---|---|
| 5/30/05 | 207 | 10:38 - 16:01 | 252-254 | 7 | $2 \times 10^3$ | 500 | 0.8 | 5 |
| 6/24/05 | 225 | 11:44 - 15:54 | 259-261 | 100 | $9 \times 10^3$ | 100 | 3 | 20 |
| 7/28/05 | 254 | 10:38 - 15:45 | 266-268 | 200 | $1 \times 10^4$ | 70 | 3 | 30 |
| 8/3/05 | 263 | 10:07 - 15:34 | 260-265 | 50 | $1 \times 10^4$ | 100 | 1 | 30 |

[a] Local time
[b] Estimated concentration of dissolved organic carbon in each sample (in units of µmol-C / L), calculated based on the measured $k'_{OH}$ value using Eq. (8).



**Table 2.** Average dissolved organic carbon concentrations in snow and ice from past reports.

| Type | Location | Time Frame | Average [DOC] (μmol-C/L) | References |
|---|---|---|---|---|
| Snow | Summit, Greenland [a] | Spring (2001) | 48 | (Grannas et al., 2004) |
| | | Summer (2001) | 47 | |
| | | Fall (2001) | 33 | |
| | | Winter (2002) | 36 | |
| | Alert, Nunavut, Canada [a] | May (2002) | 58 | (Grannas et al., 2004) |
| | | February (2002) | 17 | |
| | Barrow, Alaska | Feb – Apr (2009) | 20 | (Beine et al., 2012) |
| | Taylor Valley, Antarctica [a] | | < 8 | (Lyons et al., 2007) |
| | Alberta, Canada [b] | October (1999) | 23, 34 | (Lafreniere and Sharp, 2004) |
| Ice Cores | Col du Dome | 1850 – 1976 | 9.6 – 25.5[c] | (Legrand et al., 2007) |
| | Vostok, Antarctica | 9970 BP | 0.43 | (Preunkert et al., 2011) |
| | D47, Antarctica | 9970 BP | 0.14 | (Preunkert et al., 2011) |
| | South Pole | 1010 BP | 0.62 | (Preunkert et al., 2011) |
| | Summit, Greenland | Winter (1020) | 0.8 | (Preunkert et al., 2011) |
| | | Summer (1020) | 3 | |
| | Mt. Blanc, French Alps | Winter (1925-1936) | 3.8 | (Preunkert et al., 2011) |
| | | Summer (1925-1936) | 8.2 | |
| Sea Ice | Barrow, Alaska | | 26 | (Beine et al., 2012) |

[a] Total (dissolved and particulate) organic carbon concentration.
[b] Snowmelt from a forest (23 μmol-C/L) and meadow (34 μmol-C/L) sample.
[c] Range of [DOC] values, not the average.


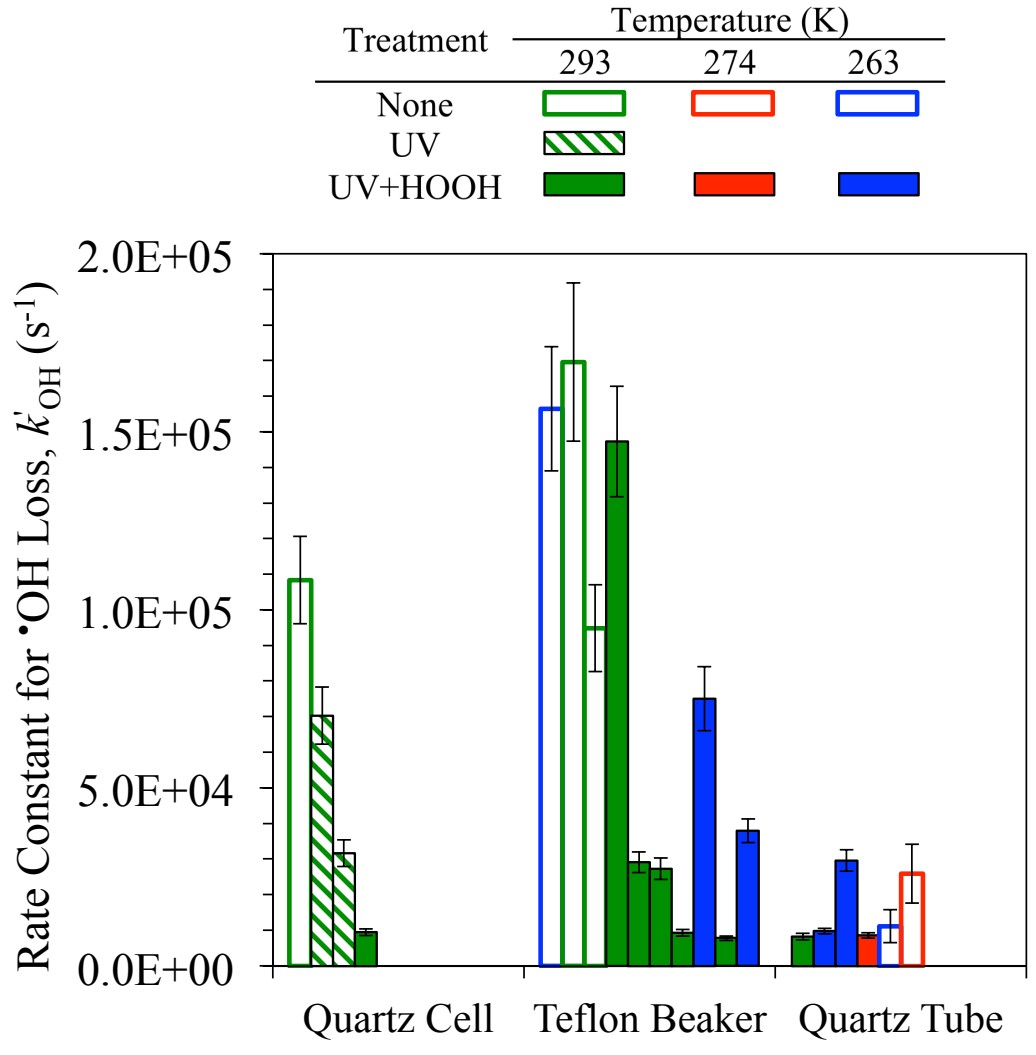

Figure 1





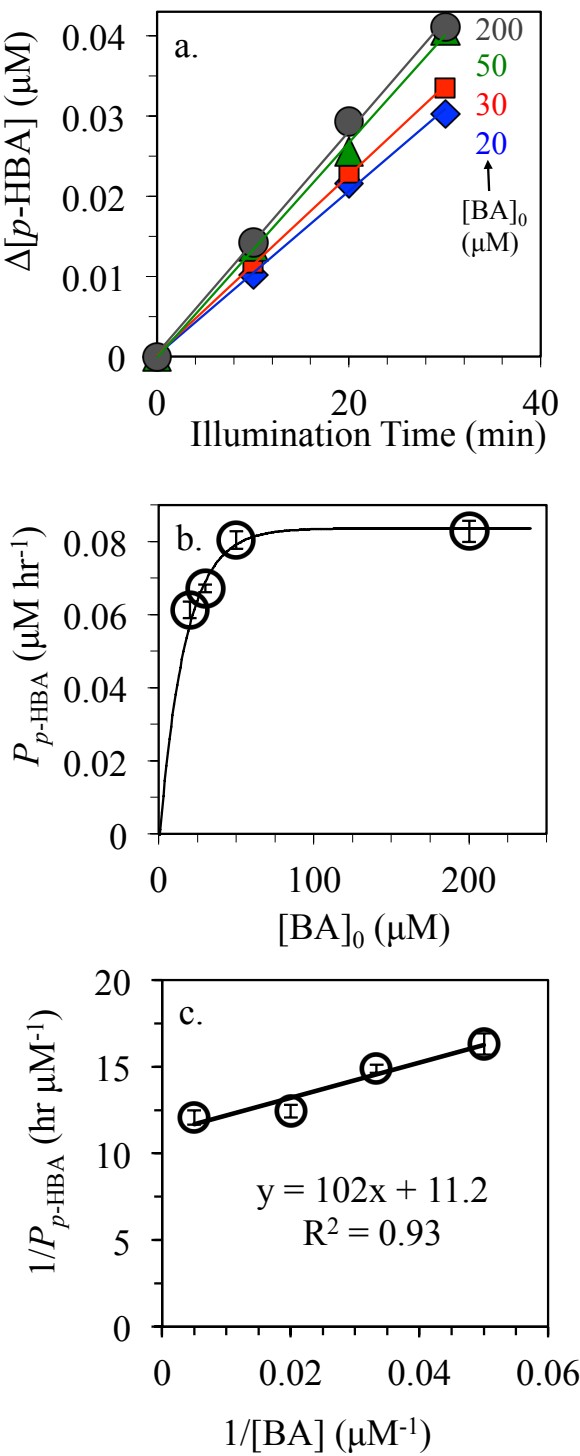

Figure 2





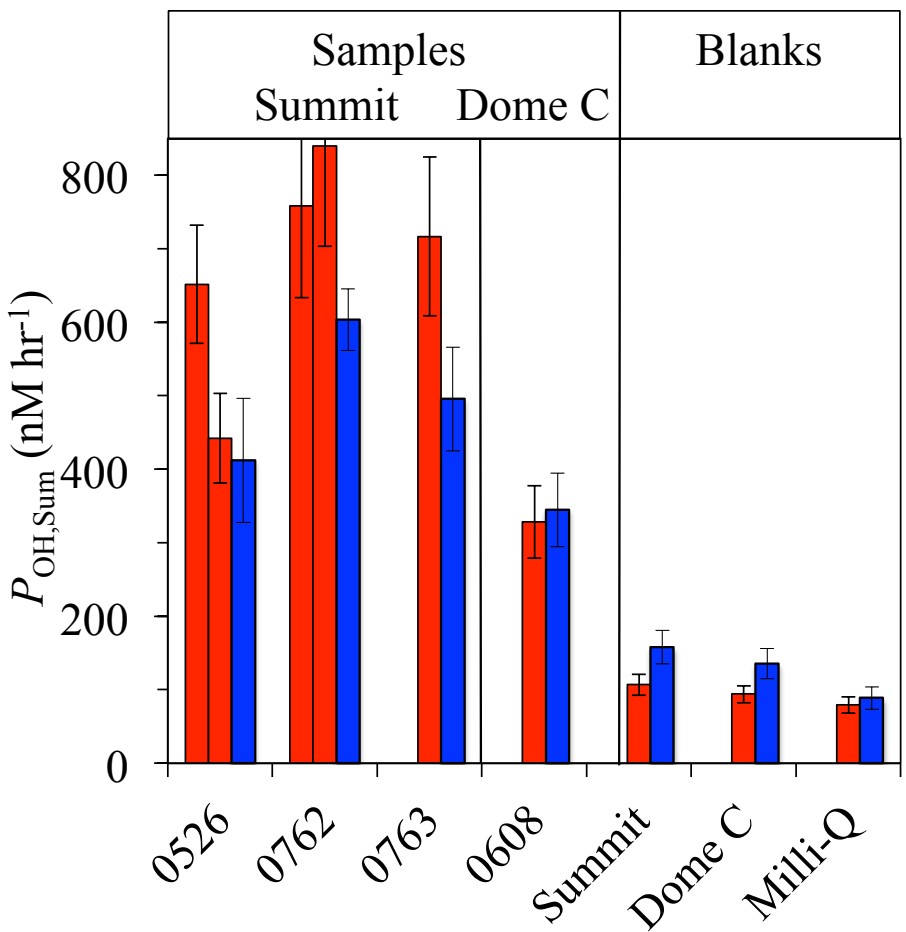

Figure 3





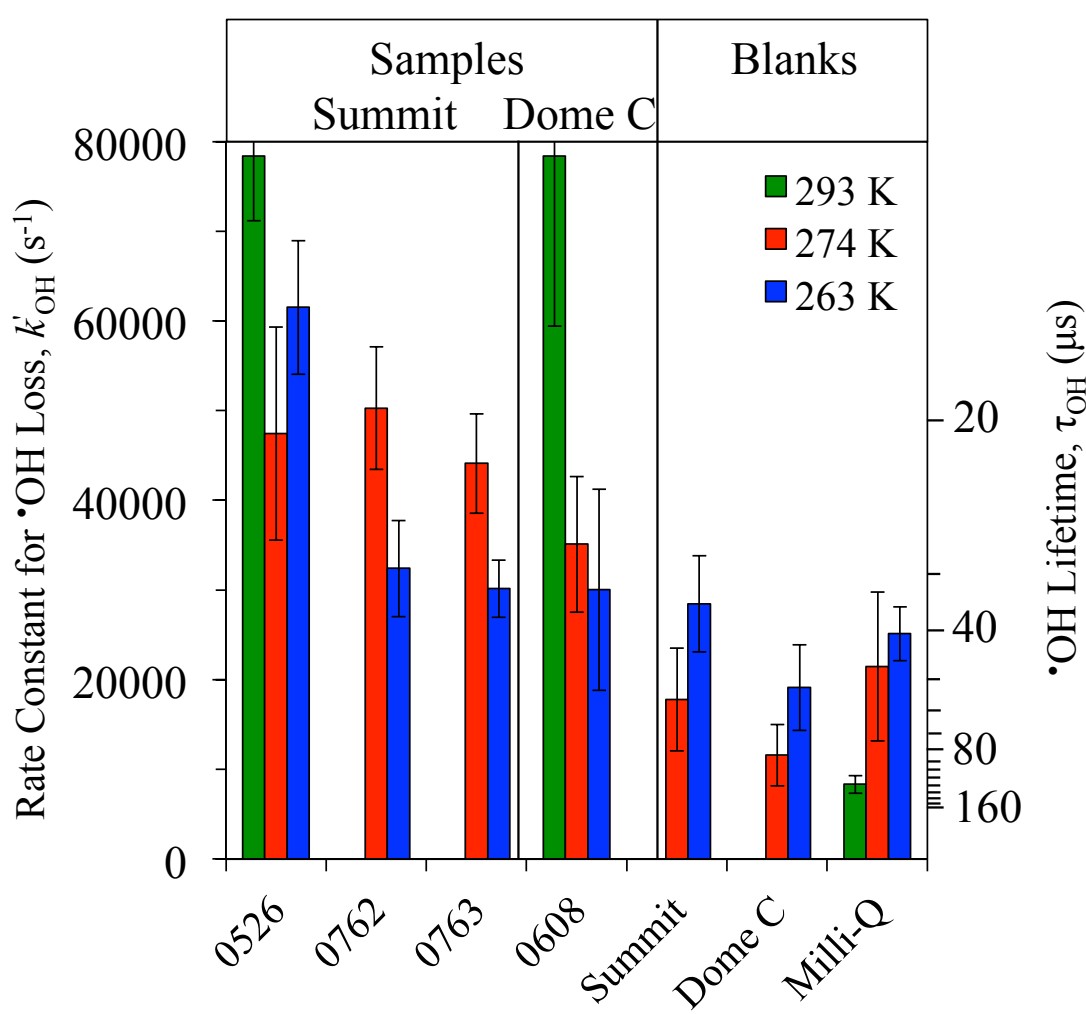

Figure 4



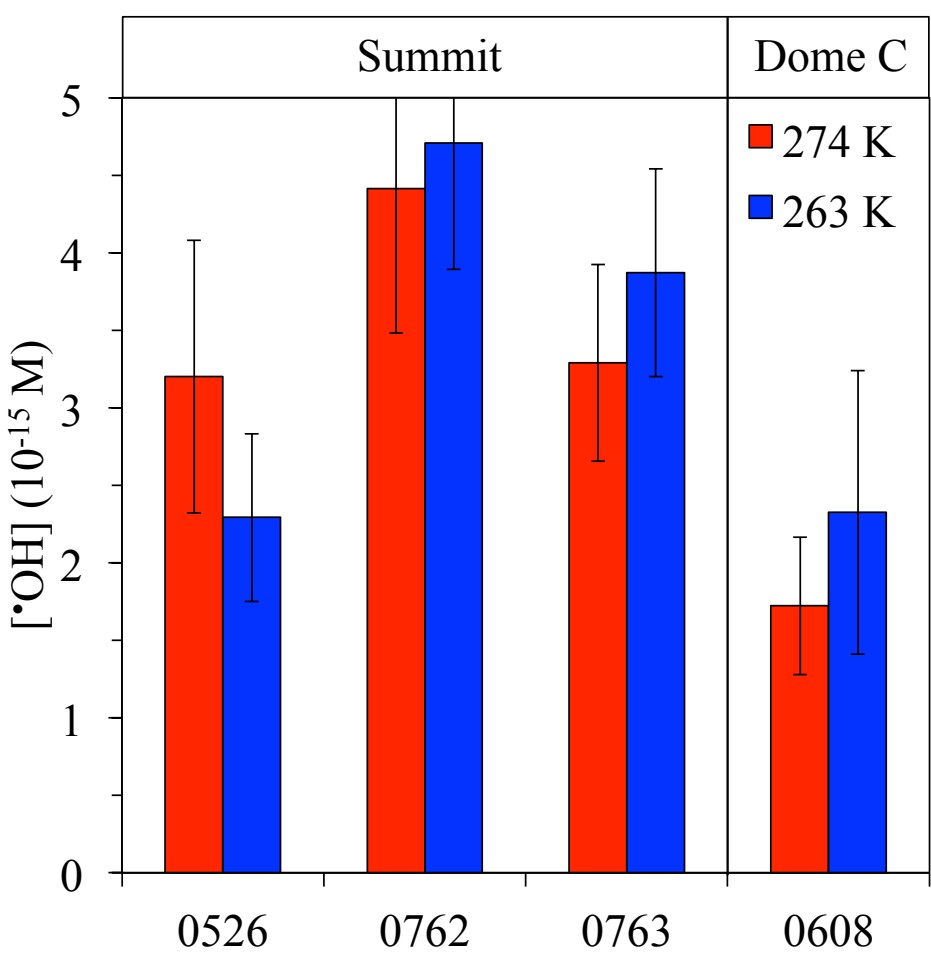

Figure 5





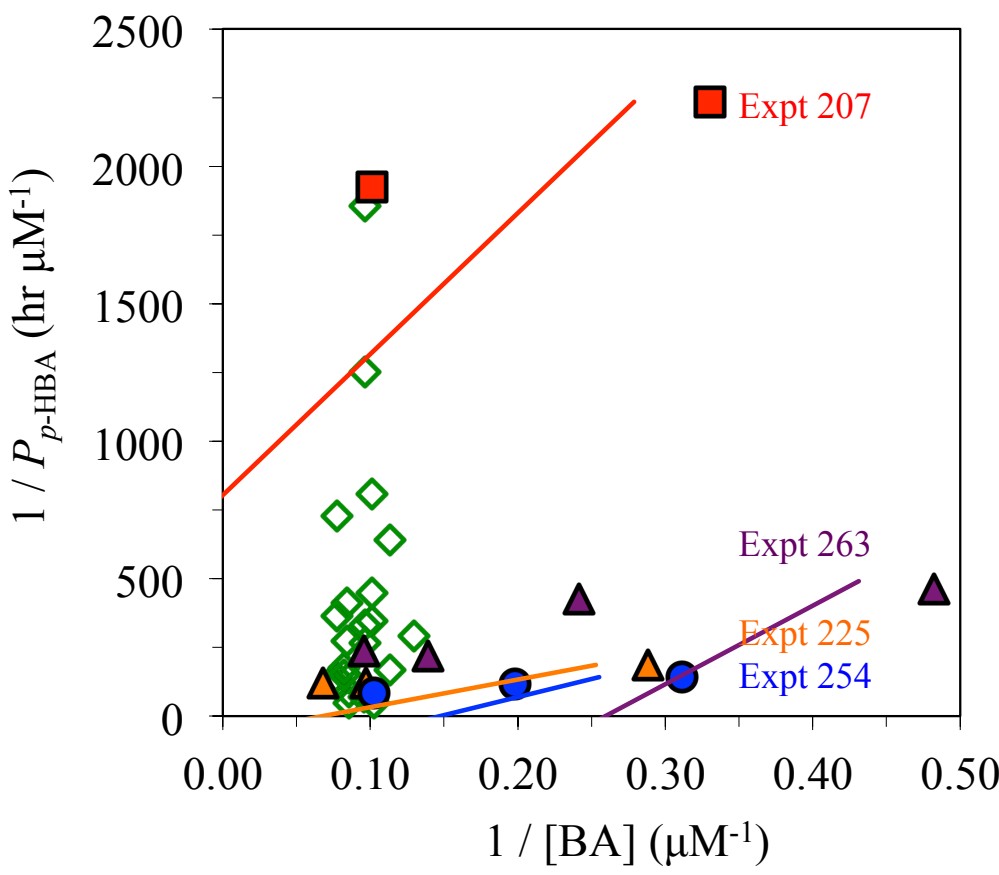

Figure 6