# Peer review of "Hydroxyl Radical in/on Illuminated Polar Snow: Formation Rates, Lifetimes, and Steady-State Concentrations"

_Atmospheric Chemistry and Physics, 2016_

## Referee Comment (RC1) · Anonymous Referee #1 · 5 Apr 2016

In this paper, Chen et al. use kinetic analysis of OH p-HBA formation in illuminated snow samples to infer OH photochemical formation rates and steady-state concentrations in (assumed) pristine snow taken from Arctic regions. This work is important and timely: in a rapidly changing and increasingly human-impacted Arctic environment, understanding the contributions to local atmospheric reactivity and oxidation state due to chemistry associated with "pristine" snow and firn air will give a yardstick against which to measure any changes due to human activities.

Although the concentrations of OH determined here are certainly in line with expectations, I do have some concerns about some of the methods and assumptions used to derive these. I outline these below.

[Figure]

First, as noted by the authors on page 15 of the MS (in the "Implications and Uncertainties" section), the experiments were all carried out by first melting the snow sample, then adding the benzoic acid, then re-freezing, carrying out the illumination, then melting again for analysis. What we know about snow reactivity (Bartels-Rausch et al. ACP (2014); Dominé et al., JPCA (2013)) indicates that morphology may indeed play a significant role in chemistry, so it could well be important here as well. As well, Kahan et al. (ACP (2010), ES&T (2010)) have shown that OH reactivity on ice surfaces may be orders of magnitude different from that within the matrix. These considerations suggest that care should be taken in trying to infer OH concentrations within a disordered layer.

Second, also as noted in the same section by the authors, the re-freezing not only alters the "natural" morphology, but could also alter the partitioning of the HOOH and organics within the sample. I think it is fair to say that we do not yet have any quantitative understanding about how different solutes distribute themselves as an aqueous solution freezes.

Third, I do worry a bit that the bulk of the MS discusses laboratory kinetic results which the authors conclude to have been significantly impacted by an unknown contaminant. What are the implications of this for the general results?

On a more technical note, I am not convinced that the slopes and intercepts obtained from 4-point fits (such as those displayed in Figure 2) are as well constrained as the authors imply. Perhaps some discussion is warranted along these lines.

Also on a technical note, I find the presentation in Section 2.7, concerning how the data were treated, to be quite confusing.

---

## Referee Comment (RC2) · Anonymous Referee #2 · 21 Apr 2016

The manuscript describes a detailed kinetic study on the photochemistry of OH in ice. The manuscript is very clear and shows that experiments were very carefully done and evaluated. The data are sound and conclusions well justified. This and relevance of the hydroxyl radical in atmospheric chemistry of polar areas clearly grants publication after a minor addition. In particular, I was impressed by the comparison of OH and O2 nicely discussing the freeze-concentration effect and how it differs for different reaction systems.

General comment on freeze-concentration effect and Liquid-Like-Regions: The term freeze-concentration effect was, to the best of my knowledge, first used to describe increasing reaction rates with the shrinking volume of the liquid fraction in a binary icesolution system. As soon as the system was completely frozen, i.e. below the eutectic, reactivity ceased. (Takenaka, N., Ueda, A., Daimon, T., Bandow, H., Dohmaru, T., and Maeda, Y. "Acceleration Mechanism of Chemical Reaction by Freezing: the Reaction of Nitrous Acid with Dissolved Oxygen" The Journal of Physical Chemistry 100, no. 32 (1996): 13874–13884. doi:10.1021/jp9525806; Takenaka, N., Ueda, A., and Maeda, Y. "Acceleration of the Rate of Nitrite Oxidation by Freezing in Aqueous-Solution" Nature 358, no. 6389 (1992): 736–738. doi:10.1038/358736a0). I recommend to stick to this terminology which implies that one either has a freeze-concentration effect AND a liquid fraction in a frozen (binary) system or a (potentially high) reactivity in a completely frozen system that might take place in LLR/in a qll/in a liquid-like brine/or on the surface. More general, I'd appreciate a discussion on the phase behaviour of your samples. As the composition is not known, this is agreeable difficult. Nevertheless, maybe reactivity is similar in ice and liquid, because the ice is actually a mixture of ice and a reactant solution. For example, if H2O2 would be the origin of OH, one might expect liquid well down to -50 °C (Foley, W. T. and Giguère, P. A. "Hydrogen Peroxide and Its Analogues: II. Phase Equilibrium in the System Hydrogen Peroxide-Water" Canadian Journal of Chemistry-Revue Canadienne De Chimie 29, no. 2 (1951): 123–132.) May I therefore suggest to include a short discussion on the possibility of the presence of liquid as reaction medium in your samples, both in the introduction and when comparing the results.

Specific comments:

P1 - 12ff: Laboratory studies show that . . .. Does the contrast to other oxidant refer to both OH kinetics and concentration? Or only to the concentration? Could you reword.

P 3 - 15: While it has not been measured experimentally, the freeze-concentration effect migh also alter . . . This is certainly a very valid hypothesis. I would suggest to underline it further by stating a few examples where reactivity was observed to change, referring to Klan and/or Donaldson earlier work (Bartels-Rausch, T., Jacobi, H.-W., Ka-han, T. F., Thomas, J. L., Thomson, E. S., Abbatt, J. P. D., Ammann, M., Blackford, J.

[Figure]

R., Bluhm, H., Boxe, C., Dominé, F., Frey, M. M., Gladich, I., Guzman, M. I., Heger, D., Huthwelker, T., Klán, P., Kuhs, W. F., Kuo, M. H., Maus, S., Moussa, S. G., McNeill, V. F., Newberg, J. T., Pettersson, J. B. C., Roeselova, M., and Sodeau, J. R. "A Review of Air–Ice Chemical and Physical Interactions (AICI): Liquids, Quasi-Liquids, and Solids in Snow" Atmospheric Chemistry and Physics 14, no. 3 (2014): 1587–1633. doi:10.5194/acp-14-1587-2014)

P 8 - 10: This is surprising as the apparent rate constant of OH towards organics spans several orders of magnitude (Schwarzenbach, R. P., Gschwend, P. M., and Imboden, D. M. "Environmental Organic Chemistry" (2005): doi:10.1002/0471649643). Would this imply that reactivity of DOC is dominated by one class of organics? Could you comment (or further support) that statement in the manuscript.

P 11 - 6: "if the blank samples….contamination Milli-Q…". This is a reasanoble assumption. Could you further support it by stating the type of Milli-Q used, does it use UV and does it maybe not filter the organics efficiently? Then one might expect high peroxide concentrations.

---

## Author Comment (AC1) · 23 Jun 2016

Referee comments are in plain text.

*Our responses are indented, in italicized text.*

**Anonymous Referee #1**

In this paper, Chen et al. use kinetic analysis of OH p-HBA formation in illuminated snow samples to infer OH photochemical formation rates and steady-state concentrations in (assumed) pristine snow taken from Arctic regions. This work is important and timely: in a rapidly changing and increasingly human-impacted Arctic environment, understanding the contributions to local atmospheric reactivity and oxidation state due to chemistry associated with "pristine" snow and firn air will give a yardstick against which to measure any changes due to human activities.

Although the concentrations of OH determined here are certainly in line with expectations, I do have some concerns about some of the methods and assumptions used to derive these. I outline these below.

First, as noted by the authors on page 15 of the MS (in the "Implications and Uncertainties" section), the experiments were all carried out by first melting the snow sample, then adding the benzoic acid, then re-freezing, carrying out the illumination, then melting again for analysis. What we know about snow reactivity (Bartels-Rausch et al. ACP (2014); Dominé et al., JPCA (2013)) indicates that morphology may indeed play a significant role in chemistry, so it could well be important here as well. As well, Kahan et al. (ACP (2010), ES&T (2010)) have shown that OH reactivity on ice surfaces may be orders of magnitude different from that within the matrix. These considerations suggest that care should be taken in trying to infer OH concentrations within a disordered layer.

> *We agree with the reviewer that sample morphology and solute location could be important factors in ice photochemistry, including in our OH experiments. We tried to use a volatile probe (benzene) to determine OH in the field at Summit without melting the snow, but benzene was too volatile to stick appreciably to the snow grains.  Thus we needed to use benzoate, which required melting and refreezing the snow.*

> *We have expanded upon the solute partitioning and solute mixing uncertainties in the revised manuscript (section 3.5).  As part of this we have added an additional sentence in to indicate that our results should be considered a "first-order estimate of hydroxyl radical kinetics in natural snow".*

> *There do not appear to be significant amounts of HOOH at the air-ice interface (QLL) in natural snow (or in our reconstituted samples), so the QLL enhancement seen by Kahan and others is likely of little relevance to OH.  We have revised the paragraph on uncertainties to more clearly state that HOOH in natural snow is probably in the bulk ice matrix, while HOOH in our reconstituted samples is probably in liquid-like regions.  (As an aside, the QLL enhancements seen by Kahan in the cited references are a factor of 4 – 5, and not the orders of magnitude stated by the reviewer.)*

Second, also as noted in the same section by the authors, the re-freezing not only alters the "natural" morphology, but could also alter the partitioning of the HOOH and organics within the sample. I think it is fair to say that we do not yet have any quantitative understanding about how different solutes distribute themselves as an aqueous solution freezes.

> *We agree and have included further discussion of HOOH location in our revised paragraph. We have some new experimental evidence that the solutes in samples such as ours will be*

*pushed into internal liquid-like regions and have cited this in-process manuscript in Cryosphere in the revised text of section 3.5.*

Third, I do worry a bit that the bulk of the MS discusses laboratory kinetic results which the authors conclude to have been significantly impacted by an unknown contaminant. What are the implications of this for the general results?

*The laboratory results have two main purposes.  One is to compare OH kinetics in solution and ice, which can't be done in the (freezing) field.  The second (initially unintended) purpose is to evaluate the approach of bringing field samples into the laboratory for study, which is commonly used since it's difficult to study snow photochemical processes in the field. Our laboratory results in the current manuscript show that such an approach is potentially perilous because of contamination.  We were quite careful in our laboratory study of OH to avoid contamination, but our results show that it was a major problem.  So while the laboratory results are not useful for understanding OH in field snow, they are a quantitative example of the contamination issues that can arise from the common practice of studying field samples in the laboratory.*

On a more technical note, I am not convinced that the slopes and intercepts obtained from 4-point fits (such as those displayed in Figure 2) are as well constrained as the authors imply. Perhaps some discussion is warranted along these lines.

*While we have used standard statistical propagation of errors, we agree with the reviewer that the resulting errors do not generally express all of the uncertainties inherent in the measurements.  We addressed this in two ways in the original manuscript: (1) the discussion of the contamination of the samples studied in the laboratory (which dwarfs the experimentally determined uncertainties) and (2) reporting our field results to only one significant figure (Table 1).  To this we have added our new statement of uncertainty in section 3.5, the statement that our results are a first-order estimate of OH kinetics.  Together we think these components give a representative picture of the overall uncertainties.*

Also on a technical note, I find the presentation in Section 2.7, concerning how the data were treated, to be quite confusing.

*We have modified Section 2.7 in several places in order to clarify data treatment and the experimental methods.*

**Anonymous Referee #2**

The manuscript describes a detailed kinetic study on the photochemistry of OH in ice. The manuscript is very clear and shows that experiments were very carefully done and evaluated. The data are sound and conclusions well justified. This and relevance of the hydroxyl radical in atmospheric chemistry of polar areas clearly grants publication after a minor addition. In particular, I was impressed by the comparison of OH and O2 nicely discussing the freeze-concentration effect and how it differs for different reaction systems.

General comment on freeze-concentration effect and Liquid-Like-Regions: The term freeze-concentration effect was, to the best of my knowledge, first used to describe increasing reaction rates with the shrinking volume of the liquid fraction in a binary ice- solution system. As soon as the system was completely frozen, i.e. below the eutectic, reactivity ceased. (Takenaka, N., Ueda, A., Daimon, T., Bandow, H., Dohmaru, T., and Maeda, Y. "Acceleration Mechanism of Chemical Reaction by Freezing: the Reaction of Nitrous Acid with Dissolved Oxygen" The Journal of Physical Chemistry 100, no. 32 (1996): 13874–13884. doi:10.1021/jp9525806; Takenaka, N., Ueda, A., and Maeda, Y. "Acceleration of the Rate of Nitrite Oxidation by Freezing in Aqueous-Solution" Nature 358, no. 6389 (1992): 736–738. doi:10.1038/358736a0). I recommend to stick to this terminology which implies that one either has a freeze-concentration effect AND a liquid fraction in a frozen (binary) system or a (potentially high) reactivity in a completely frozen system that might take place in LLR/in a qll/in a liquid-like brine/or on the surface. More general, I'd appreciate a discussion on the phase behaviour of your samples. As the composition is not known, this is agreeable difficult. Nevertheless, maybe reactivity is similar in ice and liquid, because the ice is actually a mixture of ice and a reactant solution. For example, if H2O2 would be the origin of OH, one might expect liquid well down to -50 C (Foley, W. T. and Giguère, P. A. "Hydrogen Peroxide and Its Analogues: II. Phase Equilibrium in the System Hydrogen Peroxide-Water" Canadian Journal of Chemistry-Revue Canadienne De Chimie 29, no. 2 (1951): 123–132.) May I therefore suggest to include a short discussion on the possibility of the presence of liquid as reaction medium in your samples, both in the introduction and when comparing the results.

> *We consider a sample to be completely frozen when it has reached thermodynamic equilibrium with the surrounding sub-freezing temperature. At temperatures above the eutectic, the frozen sample should contain (mostly) pure water ice, concentrated liquid-like regions (LLRs) containing the solutes, and gas bubbles. (Below the eutectic temperature all of the solutes should be in a solid, frozen phase, although we and others have found evidence for reactive LLRs even below the eutectic (Bower and Anastasio, 2013; Cho et al., 2002).*

> *The freeze-concentration factor, as defined by Takenaka (Takenaka and Bandow, 2007), is the ratio of solute concentration in the LLRs ("micropockets") compared to in the initial solution. In this same work the authors describe the LLR composition and kinetics using freezing-point depression. So our use of the terms "freeze-concentration effect" and "freeze-concentration factor", and our kinetic treatment of our data, are consistent with this work by Takenaka. We have modified the introduction to include this reference and mention that the freeze-concentration effect can also enhance the rate of thermal reactions in ice.*

> *Consistent with these ideas, our recent imaging work on laboratory samples show three phases: pure ice, gas bubbles, and liquid-like regions containing the solutes (Hullar and Anastasio, In review). HOOH is the major precursor for OH, and is a major solute in the samples, but other solutes (including $NO_3^-$, $SO_4^{2-}$, $NH_4^+$, and $Ca^{2+}$) together account for approximately half of the total solutes (Anastasio et al., 2007; Dibb et al., 2010). Thus,*

*while a HOOH-water system doesn't capture the full complexity of our refrozen snow samples, the solutes are likely primarily in concentrated LLRs.*

Specific comments:

P1 - 12ff: Laboratory studies show that . . .. Does the contrast to other oxidant refer to both OH kinetics and concentration? Or only to the concentration? Could you reword.

*We have modified the sentence to clarify our meaning.*

P 3 - 15: While it has not been measured experimentally, the freeze-concentration effect might also alter . . . This is certainly a very valid hypothesis. I would suggest to underline it further by stating a few examples where reactivity was observed to change, referring to Klan and/or Donaldson earlier work (Bartels-Rausch, T., Jacobi, H.-W., Kahan, T. F., Thomas, J. L., Thomson, E. S., Abbatt, J. P. D., Ammann, M., Blackford, J. R., Bluhm, H., Boxe, C., Dominé, F., Frey, M. M., Gladich, I., Guzman, M. I., Heger, D., Huthwelker, T., Klán, P., Kuhs, W. F., Kuo, M. H., Maus, S., Moussa, S. G., McNeill, V. F., Newberg, J. T., Pettersson, J. B. C., Roeselova, M., and Sodeau, J. R. "A Review of Air–Ice Chemical and Physical Interactions (AICI): Liquids, Quasi-Liquids, and Solids in Snow" Atmospheric Chemistry and Physics 14, no. 3 (2014): 1587–1633. doi:10.5194/acp-14-1587-2014)

*In this portion of the introduction we are referring specifically to the possibility that the concentration of hydroxyl radical might be enhanced in ice compared to in solution. Since there are no published data that address this topic (e.g., the Bartels-Rausch paper does not), we haven't added a citation. We do talk more broadly about oxidants in general earlier in the paragraph and we have added a new reference from Grannas' group (Fede and Grannas, 2015) to this portion of the text.*

P 8 - 10: This is surprising as the apparent rate constant of OH towards organics spans several orders of magnitude (Schwarzenbach, R. P., Gschwend, P. M., and Imboden, D. M. "Environmental Organic Chemistry" (2005): doi:10.1002/0471649643). Would this imply that reactivity of DOC is dominated by one class of organics? Could you comment (or further support) that statement in the manuscript.

*We were surprised by this result as well, although it turns out that the surface water community has known this for some time. As we discuss in detail in our previous work (Arakaki et al., 2013), while there are a wide range of bimolecular OH rate constants for organic compounds, when expressed in terms of carbon concentration (i.e,. in units of L mol-C$^{-1}$ s$^{-1}$, rather than in terms of the molar concentration of compound), the average OH rate constant is fairly robust over a very wide range of carbon numbers (from approximately 2 to 30). Since the organic carbon composition in natural samples is very complex, likely composed of 1000s of compounds, we suspect that the relatively robust result in natural samples is because of averaging across so many species. On the other hand, the relative standard deviation of the average rate constant is 50% for atmospheric waters, so there is still significant spread between different samples. We have added sentence about this to section 2.8 in the revised manuscript; for more information we encourage readers to see the Arakaki et al. reference.*

P 11 - 6: "if the blank samples. . ..contamination Milli-Q. . .". This is a reasanoble assumption. Could you further support it by stating the type of Milli-Q used, does it use UV and does it maybe not filter the organics efficiently? Then one might expect high peroxide concentrations.

*We defined the type of Milli-Q system in section 2.1. We have added a note that the system does not treat the water with UV radiation in section 2.1 and on page 11 in the revised manuscript.*

**References cited:**

Anastasio, C., Galbavy, E. S., Hutterli, M. A., Burkhart, J. F., and Friel, D. K.: Photoformation Of Hydroxyl Radical On Snow Grains At Summit, Greenland, Atmospheric Environment, 41, 5110-5121, 2007.

Arakaki, T., Anastasio, C., Kuroki, Y., Nakajima, H., Okada, K., Kotani, Y., Handa, D., Azechi, S., Kimura, T., Tsuhako, A., and Miyagi, Y.: A General Scavenging Rate Constant for Reaction of Hydroxyl Radical with Organic Carbon in Atmospheric Waters, Environmental Science & Technology, 47, 8196-8203, 2013.

Bower, J. P. and Anastasio, C.: Using Singlet Molecular Oxygen To Probe The Solute And Temperature Dependence Of Liquid-Like Regions In/On Ice, J Phys Chem A, 117, 6612-6621, 2013.

Cho, H., Shepson, P. B., Barrie, L. A., Cowin, J. P., and Zaveri, R.: NMR Investigation Of The Quasi-Brine Layer In Ice/Brine Mixtures, Journal of Physical Chemistry B, 106, 11226-11232, 2002.

Dibb, J. E., Ziemba, L. D., Luxford, J., and Beckman, P.: Bromide and other ions in the snow, firn air, and atmospheric boundary layer at Summit during GSHOX, Atmospheric Chemistry and Physics, 10, 9931-9942, 2010.

Fede, A. and Grannas, A. M.: Photochemical Production of Singlet Oxygen from Dissolved Organic Matter in Ice, Environmental Science & Technology, 49, 12808-12815, 2015.

Hullar, T. and Anastasio, C.: Direct visualization of solute locations in laboratory ice samples, The Cryosphere, doi: 10.5194/tc-2015-197, In review. http://dx.doi.org/10.5194/tc-2015-5197, In review.

Takenaka, N. and Bandow, H.: Chemical Kinetics Of Reactions In The Unfrozen Solution Of Ice, Journal of Physical Chemistry A, 111, 8780-8786, 2007.